# Breaking the Gingival Barrier in Periodontitis

**DOI:** 10.3390/ijms24054544

**Published:** 2023-02-25

**Authors:** Ljubomir Vitkov, Jeeshan Singh, Christine Schauer, Bernd Minnich, Jelena Krunić, Hannah Oberthaler, Sonja Gamsjaeger, Martin Herrmann, Jasmin Knopf, Matthias Hannig

**Affiliations:** 1Clinic of Operative Dentistry, Periodontology and Preventive Dentistry, Saarland University, 66421 Homburg, Germany; 2Department of Environment & Biodiversity, University of Salzburg, 5020 Salzburg, Austria; 3Department of Dental Pathology, University of East Sarajevo, 71123 East Sarajevo, Bosnia and Herzegovina; 4Department of Internal Medicine 3—Rheumatology and Immunology, Friedrich-Alexander-Universität Erlangen-Nürnberg (FAU) and Universitätsklinikum Erlangen, 91054 Erlangen, Germany; 5Deutsches Zentrum für Immuntherapie (DZI), Friedrich-Alexander-Universität Erlangen-Nürnberg and Universitätsklinikum Erlangen, 91054 Erlangen, Germany; 6Ludwig Boltzmann Institute of Osteology at Hanusch Hospital of OEGK and AUVA Trauma Centre Meidling, 1st Med Department Hanusch Hospital, 1140 Vienna, Austria

**Keywords:** mechanical damages, barrier break, tight junctions, epithelial discontinuity, neutrophils, Oncostatin M, tissue fracture, stretching

## Abstract

The break of the epithelial barrier of gingiva has been a subject of minor interest, albeit playing a key role in periodontal pathology, transitory bacteraemia, and subsequent systemic low-grade inflammation (LGI). The significance of mechanically induced bacterial translocation in gingiva (e.g., via mastication and teeth brushing) has been disregarded despite the accumulated knowledge of mechanical force effects on tight junctions (TJs) and subsequent pathology in other epithelial tissues. Transitory bacteraemia is observed as a rule in gingival inflammation, but is rarely observed in clinically healthy gingiva. This implies that TJs of inflamed gingiva deteriorate, e.g., via a surplus of lipopolysaccharide (LPS), bacterial proteases, toxins, Oncostatin M (OSM), and neutrophil proteases. The inflammation-deteriorated gingival TJs rupture when exposed to physiological mechanical forces. This rupture is characterised by bacteraemia during and briefly after mastication and teeth brushing, i.e., it appears to be a dynamic process of short duration, endowed with quick repair mechanisms. In this review, we consider the bacterial, immune, and mechanical factors responsible for the increased permeability and break of the epithelial barrier of inflamed gingiva and the subsequent translocation of both viable bacteria and bacterial LPS during physiological mechanical forces, such as mastication and teeth brushing.

## 1. Introduction

The teeth-adjacent gingiva, also termed junctional epithelium, is a unique epithelial structure providing a barrier to the tooth-adjacent microbiota. Between junctional epithelium and tooth surface, there exists a narrow furrow-like lumen termed sulcus in healthy periodontium and a deep crevice in periodontitis, both filled with the gingival crevicular fluid (GCF), which is the blood plasma transudate continuously provided by the junctional epithelium. The junctional epithelium in periodontitis is denoted as long junctional epithelium. Both gingival and gut epithelium are exposed to multitudes of bacteria, and both act as a barrier against bacteria and their metabolic products. Both gut and gingival epithelial lining are parts of the digestive tract and have very similar flora, and their penetration by bacteria and LPS results in systemic low-grade endotoxaemia and maladaptive trained immunity. In some diseases, e.g., Morbus Crohn, similar histopathology in gut and periodontium is found. As most aspects of gut pathology are studied in detail, some of them may be used as a clue for gingival pathology, which is under-investigated. Although a part of the digestive tract, gingiva essentially differs from the gut epithelium as it is multilayer, squamous, and lacks Goblet as well as Paneth cells. In general, four defence layers of the gut barrier have been recognised: (i) intestinal alkaline phosphatase, which detoxifies lipopolysaccharide (LPS); (ii) a mucus layer entrapping the bacteria; (iii) the epithelial lining; and (iv) antibacterial peptides [1]. The gingiva is deprived of the first two layers, which are characteristic of the gut barrier. Gingival epithelium does not produce intestinal alkaline phosphatase and also lacks the mucus layer. Instead, the first layer in the gingival barrier is GCF. The continuous supply of GCF washes off the epithelial surface of PAPMs. The fresh GCF supply thus forms a layer-like halo, somewhat protecting the TJs from LPS and other PAMPs. GCF contains complement, IgA, LPS binding protein, plasma alkaline protease, and epithelial defensins, so this layer also complies with the fourth gut barrier layer, i.e., the antibacterial peptides. The second layer of the gingival barrier is the neutrophil extracellular traps (NETs) [2,3,4,5]. NETs are evolutionary conserved structures of the innate immune system. These chromatin-backboned web-like meshworks are produced by activated neutrophils mainly as a response to pathogen challenge. NETs kill, harm, and entrap bacteria and prevent their dissemination [6]. The last two NET properties emulate the main properties of the mucus layer. The third and final layer of the gingival barrier is the epithelial lining. The main function of the junctional epithelium is to provide a physical, biochemical, and pathogen barrier to the outside. However, both ligature-induced periodontitis [7] and human periodontitis are characterised by transitory bacteraemia triggered via physiological mechanical strains, e.g., mastication. Oral hygiene– and chewing-induced bacteraemia in some subjects [8,9] are unquestionable signs of a break in the gingival barrier.

The topical application of lipopolysaccharides (LPS) results in the development of experimental periodontitis, as reported earlier [10,11]. The abundance of LPS within the dental biofilm accumulated on tooth ligature suffices to initiate experimental periodontitis [12]. Gram-negative bacteria predominantly compose the dental biofilms and serve as a reservoir for LPS. This abundance of LPS is considered the main etiological factor driving late-onset human periodontitis [13]. In contrast, oral commensal streptococci antagonise periodontal pathogens such as *Porphyromonas gingivalis* [14]. The resulting persistent LPS supply through gingiva is a requirement for developing low-grade systemic inflammation (LGI) and the subsequent maladaptive trained immunity (MTI) [5,15,16] as well as other systemic diseases [17,18]. Another way to achieve a LPS surplus is to increase the permeability of the junctional epithelium (Figure 1a). Taken together, the surplus of LPS penetrating the junctional epithelium appears to be responsible for gingival inflammation, subsequent bacteraemia, and, in the long run, low-grade endotoxaemia, LGI, and MTI characterised by neutrophil hyper-responsiveness [19]. The accumulation of dental biofilm and an increase in gingival permeability are the two preconditions responsible for the surplus of LPS penetrating the gingival barrier, but gingiva-dependent bacteraemia additionally relies on mechanical strains.

In this study, we aimed to consider the dysregulation of the gingival barrier and the possible mechanisms of its break, as well as reveal new perspectives with respect to studying periodontal pathology, particularly gingival permeability and barrier break.

## 2. Epithelial Barrier of the Gingiva

Sheets of connected epithelial cells cover all host surfaces exposed to the environment and build a barrier restricting the penetration of harmful substances and viable pathogens. Molecules can cross the epithelium either through cells (transcellular transport) or between cells (paracellular transport). Tight junctions (TJs) are responsible for sealing the paracellular space and discriminating among small solutes, such as ions, water, and small uncharged molecules, while generally restricting the passage of larger molecules and microorganisms [20,21]. The impermeability of gingival TJs for low-weight molecules, e.g., ruthenium red, has been demonstrated [22]. Claudins are understood to be the backbone of TJs and play a major role in a TJ’s ability to seal the paracellular space [23] (Figure 1).

TJ selectivity is regulated through claudin pores—channels formed by the extracellular domains of claudins—that conduct ions and small molecules based on size and charge. Paracellular permeability through claudin pores is commonly called the ‘pore pathway’. A second pathway, known as the ‘leak pathway’, allows larger molecules to cross TJs, albeit with less selectivity and a much lower capacity than the pore pathway [24]. It has been hypothesised that leak pathway flux occurs due to the breaking and annealing of claudin strands, which may be influenced by intermolecular associations between claudins, occludin, zonula occludens-1 (ZO-1), and the actin cytoskeleton [25,26]. The junctional epithelium with large intercellular spaces and decreased levels of E-cadherin of adherens junctions has been considered a relatively weak barrier for invading bacteria [27]. The GCF efflux also passes through the junctional epithelium via the intercellular spaces. Bacteria have been considered to translocate across the intestinal barrier only when the structural or functional integrity of the epithelial barrier is compromised, but the translocation routes remain elusive [28].

## 3. TJ Compromising via a Surplus of LPS and by Oral Pathogens

LPS transition through gut epithelium occurs exclusively via the paracellular route [29], i.e., through deteriorated TJs, which are almost identical in gut and gingival epithelium (oral mucosa is a part of gut and has nearly the same flora). Whereas TJ deteriorations in gut epithelium are well studied, little is known about gingival epithelium. Thus, knowledge concerning gut epithelium can be used as a clue for understanding the gingival TJ pathology. LPS induces epithelial barrier disruption via TLR4/Myd88 signalling, the neutrophil influx, protein leak, and E-cadherin shedding in bronchoalveolar lavage fluids in a murine model of acute lung injury [30]. Intestinal epithelial barrier disruption with altered mucosal architecture via LPS has been reported in vivo [31]. The LPS-induced barrier disruption in epithelia occurs via a rapid redistribution of zonula occludens (ZO-1) (Figure 1); reducing the expression levels of ZO-1 in TJs through apical membrane depolarization and tyrosine phosphorylation can increase the permeability of the epithelial barrier and its vulnerability to secondary bacterial infections. All LPS effects are restored with the inhibition of the Ca^2^+-activated Cl¯ channel and epithelial Na+ channel, indicating the role of the LPS-induced channel hyper-activation in the regulation of paracellular pathways [32]. Overall, LPS is a robust inflammatory inducer, which in sufficient quantities can induce by itself barrier disruption in gingiva and in lower concentrations increases epithelial permeability. Furthermore, LPS-induced capillary inflammation (Figure 2) results in neutrophil recruitment ensuring an increase in epithelial permeability. In the long term, sustained LPS transition results in LGI and subsequent MTI, which is characterised by neutrophil hyperactivity predisposing one to other systemic LGI diseases, such as diabetes, rheumatoid arthritis, atherosclerosis, and also infectious diseases, e.g., COVID-19, etc. [17,18].

Gingipains of *P. gingivalis*, including Arg- or Lys-specific cysteine proteases, have been found to specifically degrade junctional adhesion molecule-1 (JAM-1), coxsackie virus, and adenovirus receptors in the tissue model, leading to increased permeability for lipopolysaccharide, peptidoglycan, and gingipains [33,34]. Some oral pathogens, such as *P. gingivalis*, *A. actinomycetemcomitans*, and *T. denticola*, are able to dysregulate the epithelial barrier [35] and aggravate periodontal inflammation; however, their clinical role appears to be of subordinate significance, as periodontitis is possible without the involvement of these oral pathogens. The share of non-bacterial pathogens in late-onset periodontitis is exceedingly restricted. The parasite *Trichomonas tenax* also damages TJs [36], whereas the fungus *Candida albicans* mechanically pierces the epithelial cells by its hyphae without TJ impairment [37]. An involvement of helminths in periodontitis has not been reported [38].

## 4. Ageing-Related Impairment of the Epithelial Barrier

Ageing is associated with increased baseline inflammation, called inflammageing, that may contribute to frailty in geriatric populations. Inflammageing could result from a decrease in anti-inflammatory mediators, such as IL-10, or a reduction in the capacity of the epithelial barrier to exclude inflammatory antigens [39]. Ageing plays a crucial role in breaking the periodontium’s host immunity, as ageing-related deterioration of TJs facilitates the bacterial translocation in gingiva. Until 30 years of age, only chronic gingivitis has been observed; after this point, late-onset periodontitis develops [40,41,42]. Ageing is a complex process involving various mechanisms that lead to the accumulation of subcellular, cellular, and intercellular damage and other age-related deleterious changes, together representing the organisms’ ‘deleteriomes’ [43]. The role of ageing is evident from the effects of rapamycin, a drug which slows ageing and extends the lifespan of multiple organisms, on elderly mice. Even a short-term treatment with rapamycin rejuvenates the aged periodontium of elderly mice, including regeneration of periodontal bone, attenuation of gingival and periodontal bone inflammation, and restoring the oral microbiome [44]. These findings provide a geroscience strategy for potentially rejuvenating periodontal health and even reversing the periodontal bone loss in the elderly [44]. On the molecular level, the most precise biomarker of ageing is based on DNA methylation profiling and is known as the ‘epigenetic clock’ [45]. These DNA alterations reveal the perspective of maintaining the barrier function of inflamed gingival epithelial cells by inhibiting DNA methylation [46].

## 5. TJ Compromising via Oncostatin M (OSM) and Neutrophil Proteases

### 5.1. Neutrophil Recruitment and Response to PAMPs

The main signal responsible for early neutrophil recruitment in inflamed gingiva is the LPS discarded in the periodontal crevice by the dental biofilm. LPS can penetrate healthy gingival epithelium in minimal quantities [47], but it is detoxified inside of epithelium by blood proteins, blood enzymes, and neutrophil-derived enzymes [19]. However, as discussed in Section 3, Section 4, Section 5 and Section 6, compromised epithelial barrier allows larger LPS quantities to pass, which cannot be topically detoxified (Figure 2).

Thus, the topical application of LPS causes experimental periodontitis [10,11]. LPS penetrates the gingival epithelium and is sensed by endothelial cells mainly via Toll-like receptor 4 (TLR4) [48]. Thereupon, endothelial cells induce the expression of selectin and intercellular adhesion molecule 1 (ICAM-1). P-selectin engages P-selectin glycoprotein ligand-1 to activate β2-integrin and initiate the neutrophil transmigration within the first 15 min. In contrast, E-selectin engages CD44 to influence neutrophil transmigration after 15 min. Complicated, complementary, and competitive mechanisms are involved in interacting with P-/E-selectins and their ligands to promote neutrophil transmigration [49].

After their transmission out of the venules into the connective tissue, neutrophils get pulled into the cytokine gradient produced by the junctional epithelium. Upon LPS stimulation, oral keratinocytes can generate diverse pro-inflammatory cytokines and chemokines, including interleukins (IL) such as IL-1, IL-6, IL-8, and tumour necrosis factor-α (TNF-α). Gingival inflammation due to pathogenic oral bacteria, e.g., *Porphyromonas gingivalis* (*P. gingivalis*) and *Aggregatibacter actinomycetemcomitans* (*A. actinomycetemcomitans*), may induce a differentiated production of these cytokines [50]. Neutrophils from the gingival connective tissue target the crevicular lumen, which contains gingival crevicular fluid (GCF), dental biofilm, and its metabolic products. They transmigrate through the junctional epithelium and emerge onto its outer surface to encounter the bacterial challenge [5]. Within the crevicular lumen, neutrophils are primed (characterised by ROS production, delayed apoptosis, and degranulation) or activated (denoted by their ability to generate NETs) via diverse PAMPs such as the dental biofilm supernatant. TLRs are not engaged in reactive oxygen species (ROS) and NET release when stimulated with supernatant from oral pathogens [51]. The PAMP recognition appears to occur via outer membrane vesicles (OMVs) [52]. Gram-negative bacteria prevail in subgingival biofilms. Thus, the leading share of bacterial membrane vesicles from dental biofilm in periodontitis are OMVs that are heavily loaded with LPS [52]. When OMVs are endocytosed by crevicular neutrophils, they release LPS from the early endosomal compartments into the cytosol [52]. Thereby, the host is capable of TLRs-independent cytosolic recognition of LPS via murine caspase-11 [53,54]. Inflammatory caspases, namely, murine caspase-11 as well as human caspase-4 and caspase-5, serve as receptors for cytosolic LPS [55]. In humans, caspase-4 and caspase-5 activate the non-canonical cytosolic LPS/caspase-4/5/Gasdermin D pathway of NET formation, which is also peptidylarginine deiminase 4–indipendent [56].

Another possibility to trigger NET formation in the crevicular lumen without the involvement of TLRs is via cleavage of the protease-activated receptor-2 (PAR2) on neutrophils surfaces, e.g., by gingipains [33].

### 5.2. Implications of Neutrophil Infiltration on Epithelial Barrier

Neutrophil-derived OSM has been shown to be important in the first stages of epithelial repair, during which basal cells proliferate to cover the wound. OSM expression during epithelial inflammation is functionally connected to neutrophil infiltration, whereas the OSM receptor is expressed on keratinocytes [57]. Therefore, if OSM expression is transient, it will allow epithelial cells to redifferentiate back into functional epithelium during the later stages of repair. However, when neutrophils are recruited to a chronic inflammatory site, they may assume under pathogenic conditions a phenotype, which constitutively produces both OSM and granulocyte-macrophage colony-stimulating factor (GM-CSF) [58]. The GM-CSF alone could induce chronic neutrophil-derived OSM in sufficient amounts to prevent the late stage of epithelial repair, causing a long-term state of barrier dysfunction. In addition, GM-CSF would also promote the long-term survival of OSM-producing neutrophils. Thus, the production of OSM by neutrophils could participate in both ongoing epithelial repair as well as promote the epithelial barrier dysfunction, depending on the duration of OSM production [59]. Thus, it is without question that OSM plays a pivotal role in late-onset periodontitis. Indeed, the levels of periodontal tissue destruction strongly correlate with OSM levels in GCF, so OSM can be used as a biomarker of periodontal disease [60].

Although OSM mediates some inflammatory pathways, antibacterial immunity per se may not be compromised in the absence of OSM activity [61]. Thus, untreated OSM−/− mice are viable and healthy, which suggests that a therapeutic blockade of OSM might cause minimal side effects. In an animal model of colitis, OSM−/− mice display reduced colon pathology, according to colonoscopy and histological assessment, when compared to wild-type controls [62]. This finding indicates the role of OSM surplus in the inflammatory epithelial pathology. Some OMVs of Gram-negative oral pathogens, e.g., *T. denticola*, induce the OSM release via an unknown signalling pathway. Actinomycin D-untreated neutrophils release much more OSM than actinomycin D-treated ones, a finding suggesting that neutrophils release granules containing OSM, an independent process that does not rely on transcription and de novo protein synthesis. Nevertheless, OSM gene transcription increases 0.4-fold in *T. denticola*–treated neutrophils. Interestingly, oral neutrophils from patients with periodontal disease have 0.24-fold more OSM transcript than healthy individuals [63].

Further, neutrophils mediate tissue damage via releasing cytokines, proteases, and other factors contained in their cytoplasmic granules and regulating the activity of the adaptive immune response, including both B cell and T cell activation [64]. Abundant crevicular neutrophils and NETs [2] overload the pocket epithelium with neutrophil-derived proteases [65,66,67]. NET-derived components such as histones [68,69,70,71] and myeloperoxidase [68] are cytotoxic to epithelial cells; neutrophil proteases damage and even kill epithelial cells as well as promote tissue damage [72,73]. High NET levels reportedly suppress keratinocyte proliferation, delay wound closure [74,75], and chronify ulcers.

Local administration of a neutrophil elastase inhibitor in a ligature induced murine model of periodontitis significantly decreased neutrophil elastase activity in periodontal tissue and attenuated periodontal bone loss. Furthermore, the transcription of proinflammatory cytokines in the gingiva, which were significantly upregulated in the periodontitis model, is downregulated by the administration of neutrophil elastase inhibitor [76]. Neutrophil elastase is known to cleave in vitro cell adhesion molecules, such as desmoglein-1, occludin, and E-cadherin, and induce exfoliation of the epithelial keratinous layer in three-dimensional human oral epithelial tissue models. The gingival permeability for fluorescein-5-isothiocyanate (FITC)-dextran or periodontal pathogens in vitro has been observed to increase by neutrophil elastase treatment of the human gingival epithelial monolayer. These findings suggest that neutrophil elastase may induce the disruption of the gingival epithelial barrier and bacterial translocation [76].

## 6. Mechanical Rupture of the Epithelial Barrier in the Gingiva

Deteriorating the TJ and adherens junction resistance via topical pathogenic factors (Section 2, Section 3, Section 4 and Section 5) appears insufficient to induce transitory gingiva-related bacteraemia but is the precondition for it. Gingiva-related transitory bacteraemia has never been reported in the absence of mechanical strain on the gingiva; consequently, the mechanical component is required to enable the bacterial translocation and subsequent bacteraemia. Indeed, bacteraemia in clinically orally healthy subjects and patients with gingivitis, as well as periodontitis, has been reported only after mastication, interdental tooth brushing, tooth brushing, and dentists’ manipulations [8,9,77]. Due to mastication and/or teeth brushing, transitory bacteraemia provides clues for understanding how viable bacteria pass the junctional epithelium. The bacteraemia ceases within a few minutes after ending the mastication or tooth brushing. This circumstance is an indication that the physiological mechanical strains are the trigger and conditio sine qua non of bacteraemia due to inflamed gingiva. The basic unit of TJs is the claudin-based strand. Mechanical changes in junctional or tissue tension can also alter the strand network morphology by reorienting the strand network or causing strand breaks [78] (Figure 3a). TJs play a major role in maintaining the integrity and impermeability of the epithelial barrier and hence act as an ideal target for pathogens to promote their translocation through the mucosa and to invade their host [79]. When TJs are inflammation-deteriorated, their rupture due to physiological mechanical strains is a convincing explanation of how viable bacteria penetrate junctional epithelium in high numbers in a short time-lapse. After the closure of a TJ break, which may be a quick process based on apical membrane-anchored serine proteinases [80], the bacteria reach the circulation via the vena cava superior and are cleared by Kupffer cells, which are the resident intravascular phagocyte population of the liver and are critical to the capture and killing of bacteria [81].

The rupture of TJs and subsequent inflammation has been reported in different tissues, an indication that ruptures of TJs are a common phenomenon in epithelial pathology. Applying hydrostatic intraductal pressures of 100 and 150 mm Hg for 10 min consistently induces pancreatic inflammation and loss of tight junction integrity in a mouse model [83]. Extra pressure causes the degradation of occludin, ZO-1, and claudin-18 in primary human small airway epithelial cells. This degradation results in a decrease of the transepithelial electrical resistance and an increase in cell layer permeability [84]. Mechanical forces during lactation give rise to breast inflammation, in which very high intra-alveolar and intra-ductal pressures are hypothesized to strain or rupture the TJs between lactocytes and ductal epithelial cells, triggering inflammatory cascades [85]. Masticatory pressure on gingiva at 73.5 mm Hg is compatible, but at 147 mm Hg bone resorption takes place, although the mechanism of bone resorption in this case remains unclear [86]. These findings suggest that non-endangering pressure on inflamed gingiva is under 147 mm Hg, i.e., no more than the normal systolic blood pressure or even below. In already inflamed tissue, the non-harming pressure range is much smaller than that of non-inflamed tissue. From a mechanobiological point of view, gingival bleeding on probing (BOP) and teeth brushing parallels the tissue squeeze [87] accompanied with petechial haemorrhages. The intermittent masticatory and brush pressure causes tensile forces in the same way as when a cutlet is beaten flat. On a cellular level, this phenomenon may be considered a living tissue fracture [82] (Figure 3b). Thus, physiological mechanical damage due to mastication and gingival rubbing induce inflammation. The latter is primarily characterised by IL-6 production by epithelial cells and an increase in gingival Th17 cells. They are associated with the further recruitment of neutrophils, which are required to clear acute bacterial infections [88,89], but in chronic infection are a source of surplus OSM.

Furthermore, the food bolus and the toothbrush exerts intermittent pressure onto the oral gingiva, whereby the content of crevicular lumen, i.e., GCF and the dispersed bacteria within, is pressed towards the pocket epithelium. OMVs, bacterial fragments, and viable bacteria might be deeply inserted through ruptured TJs, micro-wounds, and ulcers into sub-epithelial tissues, and even reach the ruptured venules. The high tendency of the venules to bleed in periodontitis and the transitory bacteremia during mastication [90], tooth brushing, and flossing [8,91,92], indicate inflamed gingiva as a portal of entry for pocket bacteria. Junctional epithelium fractures and wounds, which histologically are not confidently distinguishable from ulcers, may also be a consequence of mechanical pressure [82]. In addition, the periodontal pocket under masticatory pressure may be considered a pump, which presses bacteria and their metabolic products via the paracellular route into gingival tissues, possibly causing a GCF reflux [19]. Bacteria within GCF are pushed by intermittent masticatory and brush pressure into the gingival venules and reach blood circulation via the vena cava superior, producing both periodontitis-related bacteremia and endotoxemia.

Taken together, tissue fractures of inflamed gingival epithelium, wounds, and ulcers may enable bacterial circumvention of the gingival barrier. Indeed, pocket epithelium ulcers have been described both in late-set periodontitis [93] and experimental periodontitis [94,95], but have been rarely reported in humans with periodontitis [96,97]. The pocket epithelium amounts to few cell layers [97,98], a circumstance facilitating its mechanical rupture (wounding). Bleeding due to cyclic low mechanical pressure on the gingiva, as is frequently perceptible during teeth brushing, is an unmistakable sign of gingival wounding. The mechanism of wound generation in inflamed gingival epithelium might be similar to or even identical with a living tissue fracture [82]. The latter can be considered a further development of the barrier rupture in the epithelium. Gingiva regenerates exceptionally quickly. Oral mucosa epithelium wounds heal with a velocity of about 1 mm per day [99], i.e., about 40 µm per h, so (long) junctional epithelium wounds with a diameter of 10 µm may heal in 15 min. Thus, (long) junctional epithelium wounds with a diameter of 10 µm or less, due to physiological mechanical damages, may be efficiently non-diagnosable by histological examination. Furthermore, the junctional epithelium has a higher turnover rate than the remaining oral epithelium [100], so its restoring time is even shorter. This indicates the necessity to consider the mechanical fracturing (wounding) of junctional epithelium a dynamic, time-limited process and not a state of being.

## 7. State of Knowledge and Perspectives

The rupture of the gingival barrier is apparent from blood cultures and PCR tests of blood samples taken during or briefly after physiological mechanical strains on the gingiva. Clinically, the rupture of the gingival barrier is evident as bleeding on probing (BOP) and teeth bushing. The rupture of the gingival barrier appears to be a dynamic process of brief duration, as mastication-induced and teeth brushing–induced bacteraemia disappears in a few minutes after ceasing the physiological mechanical strain. This quick bacteraemia ceasing might be explained by the newly reported fast TJ repair mechanism based on apical membrane-anchored serine proteinases [80]. As TJ breaks are not histologically demonstrable, routine histological examinations might not be helpful. This does not exclude the possibility of using some histological methods adapted for the examination of epithelial permeability. Intravital imaging for this purpose is questionable due to the necessity of tissue deformation during an examination. Methods applied for permeability examination of other epithelial barriers (e.g., in vitro, in vivo, of gut, lung alveoli, etc.) and non-harming clinical tests have to be adapted for gingival examinations. The possibility to alleviate the deteriorating effects of LPS by using metabolites of symbiotic flora [101,102], phyto-extracts [103], and well-tolerated drugs [104] might reveal a new approach in periodontitis prophylactics.

## 8. Conclusions

Transitory bacteraemia of gingival origin is characteristic for inflamed gingiva. Many inflammatory factors, such as a surplus of LPS, OSM, neutrophil proteases, bacterial proteases, and toxins, deteriorate gingival TJs. The fact that transitory bacteraemia has been generally reported in inflamed gingiva but only in mastication, tooth brushing, and dentists’ manipulations suggests that inflammation-deteriorated TJs and adherens junctions rupture upon exposure to physiological mechanical strains. This phenomenon is also known as epithelial barrier break and, in its extreme form, as living tissue fracture. They enable the translocation of viable bacteria into blood circulation. Examination of mechanically induced rupture of the epithelial barrier in inflamed gingiva may reveal new perspectives in periodontal pathology.

## Figures and Tables

**Figure 1 ijms-24-04544-f001:**
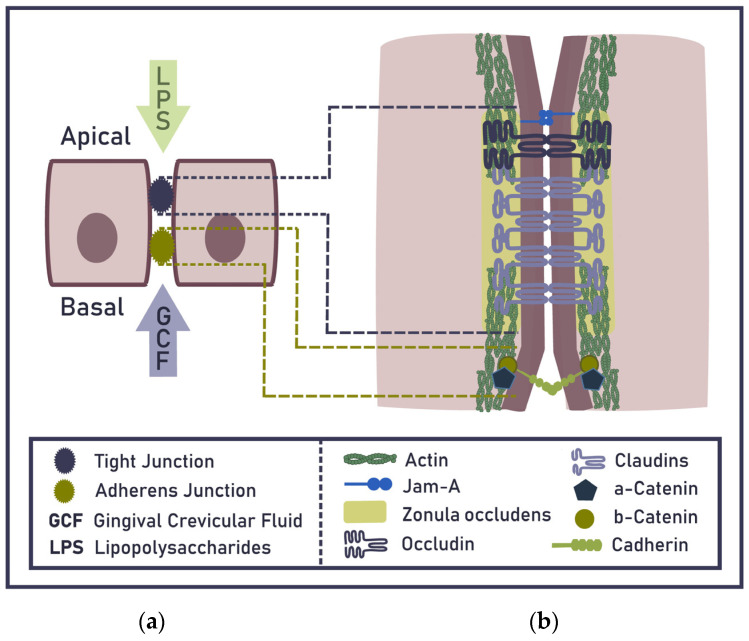
Diagrammatic representation of gingival epithelial barrier: (**a**) tight junctions (TJs) and adherens junctions providing a barrier function to the epithelium, sealing the paracellular space and the flux of gingival crevicular fluid (GCF) and the deposition of lipopolysaccharide (LPS); (**b**) a detailed scheme of TJ consisting of a major transmembrane proteins junction adhesion molecule—A (JAM-A), zonula occludens (ZO), occludins and claudins, cytoplasmic proteins a-catenin and b-catenin complexed with cadherin.

**Figure 2 ijms-24-04544-f002:**
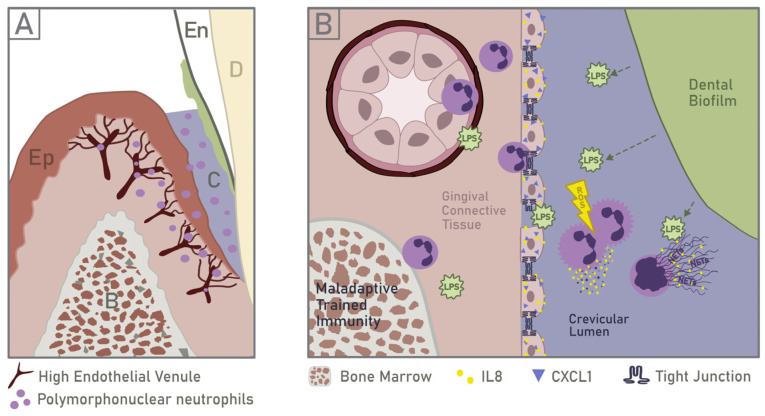
Gingival barrier dysfunction through topical factors: (**A**) a scheme of the periodontium in periodontitis; Ep—epithelium, En—tooth enamel, B—bone and bone marrow, C—crevice, D—dentin; (**B**) neutrophil recruiting and TJ disruption via LPS, neutrophil and bacterial proteases, bacterial toxins, and OSM.

**Figure 3 ijms-24-04544-f003:**
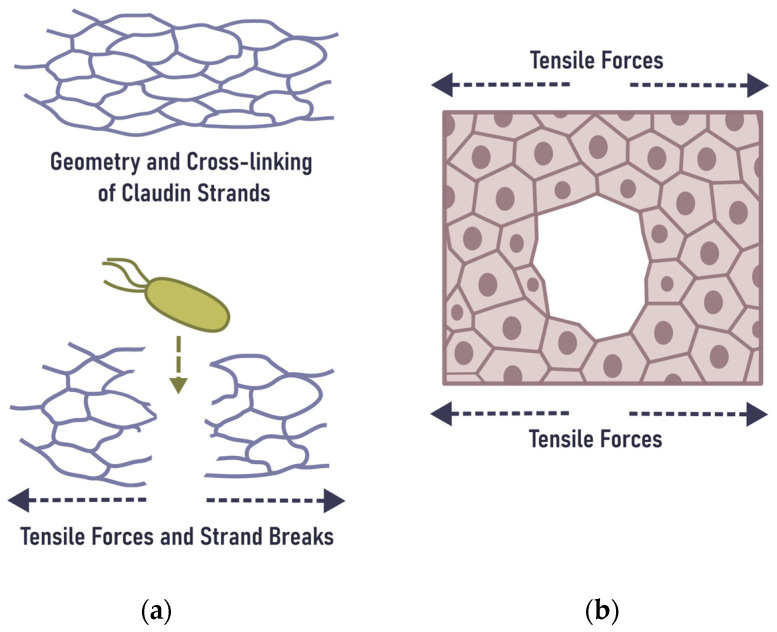
Epithelial barrier rupture due to tensile forces: (**a**) at the top—claudin strands in idle state, below—claudin strand breakage due to tensile forces and opening a temporary doorway for bacterial translocation; (**b**) a tissue-level fracture in response to tensile forces results in cell detachment, i.e., wound development, which arises from molecular scale ruptures. At the molecular scale, rupture occurs either intracellularly at connections between adhesion complexes and the cytoskeleton or extracellularly as the result of the separation of the ectodomain of intercellular adhesion proteins. Modified after Bonfanti et al. [82].

## Data Availability

Not applicable.

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
