# Peer review of "Breaking the Gingival Barrier in Periodontitis"

_ijms, 2023, doi:10.3390/ijms24054544_

Round 1
Reviewer 1 Report
In present study Vitkov et al. wrote a review on “Breaking the gingival barrier in periodontitis” where they explained the role of LPS producing biofilms in breaking the gingival barrier in periodontitis. Also they explained in detail the immune response with regard to OSM, and mechanical rupture. Few of my concerns are
1. In line 56 “Gingival epithelium does not produce intestinal alkaline phosphatase and also lacks the mucus layer” pl check the term intestinal alkaline phosphatase, how Gingival epithelium will produce intestinal enzyme.
2. Line 58- what is the role of plasma alkaline protease, and how it is detrimental to the overall gingival barrier health. What is the difference in gut alkaline phosphatase and plasma alkaline protease in relation to gingival health?
3. Line 58-59 “In addition, this layer also comprises the fourth gut layer”, clear the meaning of the line since in gingival epithelium, fourth gut layer will not be present.
4. Line 53-57- select appropriate words to explain. Here mentioned that third layer of gut “epithelial lining” is actually first layer the gingival barrier that is GCF. Here lining may not be compared with fluid, authors should rectify it.
5. How the sustenance of Gingival epithelial barrier is regulated by beneficial and harmful microbes?
6. Line 74 -75. Explain the role of gram positive bacteria also in periodontitis
7. Line 146- Ageing-related impairment of the epithelial barrier has been discussed, however in other pathological conditions also the like HIV, diabetes, radiation therapy etc. epithelial barrier is impaired. Please discuss that also.
8. Authors also should explain the direct and indirect effects of fungi like candida, and parasitic like Trichomonas tenax and helminthic infections in breaking the gingival barrier in periodontitis.
9. Suggest some pro and prebiotics that might help in preventing periodontitis.
Author Response
Our answers to the reviewers are underlined. All corrections of manuscript were highlighted.
Reviewer 1
- In line 56 “Gingival epithelium does not produce intestinal alkaline phosphatase and also lacks the mucus layer” pl check the term intestinal alkaline phosphatase, how Gingival epithelium will produce intestinal enzyme. The main function of either intestinal or plasma alkaline phosphatase is dephosphatation of LPS. Dephosphated LPS is biological inactive, i.e. does not bind TLR4 etc. In gut, intestinal alkaline phosphatase is denoted as a “defence (or barrier) layer”, in periodontium, plasma alkaline phosphatase is supplied by GCF, which is a blood plasma transudate (already mentioned in manuscript).
- Line 58- what is the role of plasma alkaline protease, and how it is detrimental to the overall gingival barrier health. What is the difference in gut alkaline phosphatase and plasma alkaline protease in relation to gingival health? Difference - produced either by gut epithelium (encoded by ALPI gene) or in the second case a component of the blood plasma (tissue-nonspecific, encoded by ALPL gene). Role - dephosphatation of LPS and thereby detoxification of LPS (mentioned at line 55).
- Line 58-59 “In addition, this layer also comprises the fourth gut layer”, clear the meaning of the line since in gingival epithelium, fourth gut layer will not be present. It was typo, not “comprises” but “complies”.
- Line 53-57- select appropriate words to explain. Here mentioned that third layer of gut “epithelial lining” is actually first layer the gingival barrier that is GCF. Here lining may not be compared with fluid, authors should rectify it. We meant “layer” as “a defence (or barrier) layer”, as used in the concerned literature DOI: 10.1210/jendso/bvz039. The continuous supply of GCF washes off the epithelial surface of PAPMs. So, the fresh GCF supply forms a layer-like halo, somewhat protecting the TJs from LPS and other PAMPs. Corresponding supplementations were inserted at Lines 54-63.
- How the sustenance of Gingival epithelial barrier is regulated by beneficial and harmful microbes? Oral commensal streptococci antagonise periodontal pathogens such as Porphyromonas gingivalis. DOI: 10.1128/jb.00257-22
- Line 74 -75. Explain the role of gram positive bacteria also in periodontitis The same as in the former point, inserted in line 81.
- Line 146- Ageing-related impairment of the epithelial barrier has been discussed, however in other pathological conditions also the like HIV, diabetes, radiation therapy etc. epithelial barrier is impaired. Please discuss that also. Periodontitis with late-onset is regarded as a predisposition and even trigger of all other LGI diseases. In Line 147 was inserted “In the long term, the sustained LPS transition results in LGI and subsequent MTI, which is characterised by neutrophil hyperactivity predisposing to other systemic LGI diseases, such as diabetes, rheumatoid arthritis, lupus erythematodes, atherosclerosis, and infectious disease e.g. COVID-19 {DOI: 10.3389/fimmu.2022.915081; DOI: 10.3389/fimmu.2022.872695}. The role of MTI in HIV remains unclear, so we cannot regard this relationship.
- Authors also should explain the direct and indirect effects of fungi like candida, and parasitic like Trichomonas tenax and helminthic infections in breaking the gingival barrier in periodontitis. Accordingly to the suggestion, we added explanations at Line 158.
- Suggest some pro and prebiotics that might help in preventing periodontitis. Despite the efficiency of pro and prebiotics to supress periodontal pathogens in vitro, the ingested or topically applied pro and prebiotics do not penetrate into periodontal crevice, as they are floated by GCF and ultimately swallowed. Also no essential clinical benefit has been reported. The clinical use of pro and prebiotics for preventing periodontitis remains questionable.
Reviewer 2 Report
This review reminds us that the role of gingival epithelial in periodontal pathology deserves more attention. It introduced some related knowledge, which makes us know more about the breaking mechanism of the gingival barrier. However, the overall organization was poor, and the narrative logic should be clearer. For example:
1. “Abstract”: makes me a bit confused. What’s the point of this review? You repeatedly mentioned transitory bacteraemia and mechanical forces (lines 24, 25, 30, 31), then said that it is “a dynamic process of short duration, endowed with quick repair mechanisms”. So, are mechanical factors responsible for barrier breaking on earth or not? How is “transitory bacteraemia” related to periodontitis (which is your theme)?
2. In the second paragraph of “the introduction, " the authors compared the gingival and gut barriers. What is the connection with the theme?
3. In the abstract, you mentioned bacterial, immune, and mechanical factors responsible for the increased permeability and break of the epithelial barrier. However, the main context seemed not to be described from these aspects respectively. Line 123: “LPS transition through gut epithelium occurs exclusively via the paracellular route”. How is it related to your theme? Why is gut epithelium mentioned here?
4. “Ageing-related impairment” belonged to which factor?
5. Line 168, another kind of LPS? Why described separately (from part 3)?
6. Line 301, “Rupture of TJs and subsequent inflammation has been reported in different tissues”. Is it related to your theme?
Author Response
Our answers to the reviewers are underlined. All corrections of manuscript were highlighted.
Reviewer 2
- “Abstract”: makes me a bit confused. What’s the point of this review? You repeatedly mentioned transitory bacteraemia and mechanical forces (lines 24, 25, 30, 31), then said that it is “a dynamic process of short duration, endowed with quick repair mechanisms”. So, are mechanical factors responsible for barrier breaking on earth or not? How is “transitory bacteraemia” related to periodontitis (which is your theme)? Sorry, it was a clumsy formulation, we meant “bacterial translocation”. Indeed, the mechanism of bacterial translocation in periodontitis has been not discussed up to now. Accordingly, we replaced “bacteraemia” with “bacterial translocation” in the abstract Line 27 in order to highlight this aspect, as the bacterial translocation being a consequence of epithelial rupture is the theme of the manuscript. (Bacteraemia is a consequence of bacteria translocation).
- In the second paragraph of “the introduction, " the authors compared the gingival and gut barriers. What is the connection with the theme? Both gut and gingival epithelial lining are parts of the digestion tract, have very similar flora, and their penetration by bacteria and LPS results in systemic low-grade endotoxaemia and maladaptive trained immunity. In some diseases, e.g. Morbus Crohn, similar histopathology is found. Indeed, the mechanisms of penetration of epithelial barrier in gut and gingiva differ. As most aspects of gut pathology are studied in detail, some of them may be used as a clue for gingival pathology, which is under-investigated.
- In the abstract, you mentioned bacterial, immune, and mechanical factors responsible for the increased permeability and break of the epithelial barrier. However, the main context seemed not to be described from these aspects respectively. Line 123: “LPS transition through gut epithelium occurs exclusively via the paracellular route”. How is it related to your theme? Why is gut epithelium mentioned here? In both cases, LPS passes paracellularly through deteriorated TJs, which are almost identical in gut and gingival epithelium (oral mucosa is a part of gut and has nearly the same flora). Whereas TJ deteriorations in gut epithelium are well studied, few is known about gingival epithelium. So, knowledge concerning gut epithelium can be used as a clue for understanding the gingival TJ pathology.
- “Ageing-related impairment” belonged to which factor? Ageing-related deterioration of TJs facilitates the bacteria translocation in gingiva.
- Line 168, another kind of LPS? Why described separately (from part 3)? – Not, but two different effects of LPS. In §3 - the direct TJs damage in epithelium by LPS were discussed, in § 5 - the LPS effects on endothelium. LPS-activated endothelial cells recruit neutrophils into connective tissue and this frequently leads to TJs damage via OSM and neutrophil proteases.
- Line 301, “Rupture of TJs and subsequent inflammation has been reported in different tissues”. Is it related to your theme? The clue is that rupture of TJs is a common phenomenon in the epithelial pathology.
Only questions have been asked by reviewer 2, but no suggestions for corrections or supplementations of the manuscript have been made. So we answered the questions and made only petit corrections of the manuscript.
Round 2
Reviewer 1 Report
Authors addressed all the comments I mentioned. I propose acceptance of the MS.
Thanks
Author Response
Many thanks for your help!
Reviewer 2 Report
I agree with the author's reply. The author should explain it in the article to make the narrative logic more clear.
Author Response
The corrections in the abstract were already inserted in the previous version. In this revision, the remaining explanations were incorporated and highlighted by “Track Changes”.
Many thanks for your help!